**Data Availability Statement:** All relevant data are within the manuscript and its Supporting Information files.

# Multiparametric MRI for assessment of early response to neoadjuvant sunitinib in renal cell carcinoma

**Stephan Ursprung**[1,2☉], **Andrew N. Priest**[1,3☉], **Fulvio Zaccagna**[1], **Wendi Qian**[1,4], **Andrea Machin**[1,4], **Grant D. Stewart**[1,2,3], **Anne Y. Warren**[1,2,3], **Timothy Eisen**[1,2,3], **Sarah J. Welsh**[1,2,3], **Ferdia A. Gallagher**[1,2], **Tristan Barrett**[1,2,3]*

**1** University of Cambridge, Cambridge, United Kingdom, **2** Cancer Research UK Cambridge Centre, University of Cambridge, Cambridge, United Kingdom, **3** Addenbrooke's Hospital, Cambridge University Hospital NHS Foundation Trust, Cambridge, United Kingdom, **4** Cambridge Cancer Trial Centre, Cancer Research UK Cambridge Institute, University of Cambridge, Cambridge, United Kingdom

☉ These authors contributed equally to this work.

* tb507@cam.ac.uk

## Abstract

### Purpose

To detect early response to sunitinib treatment in metastatic clear cell renal cancer (mRCC) using multiparametric MRI.

### Method

Participants with mRCC undergoing pre-surgical sunitinib therapy in the prospective Neo-Sun clinical trial (EudraCtNo: 2005-004502-82) were imaged before starting treatment, and after 12 days of sunitinib therapy using morphological MRI sequences, advanced diffusion-weighted imaging, measurements of $R_2$* (related to hypoxia) and dynamic contrast-enhanced imaging. Following nephrectomy, participants continued treatment and were followed-up with contrast-enhanced CT. Changes in imaging parameters before and after sunitinib were assessed with the non-parametric Wilcoxon signed-rank test and the log-rank test was used to assess effects on survival.

### Results

12 participants fulfilled the inclusion criteria. After 12 days, the solid and necrotic tumor volumes decreased by 28% and 17%, respectively (p = 0.04). However, tumor-volume reduction did not correlate with progression-free or overall survival (PFS/OS). Sunitinib therapy resulted in a reduction in median solid tumor diffusivity $D$ from $1298 \times 10^{-6}$ to $1200 \times 10^{-6} mm^2/s$ (p = 0.03); a larger decrease was associated with a better RECIST response (p = 0.02) and longer PFS (p = 0.03) on the log-rank test. An increase in R2* from 19 to $28s^{-1}$ (p = 0.001) was observed, paralleled by a decrease in $K^{trans}$ from 0.415 to $0.305min^{-1}$ (p = 0.01) and a decrease in perfusion fraction from 0.34 to 0.19 (p<0.001).

**Funding:** The project was supported by Cancer Research UK (CRUK), the CRUK Cambridge Centre, Cambridge Trust, the CRUK & Engineering and Physical Sciences Research Council Cancer Imaging Centre in Cambridge and Manchester, the Mark Foundation for Cancer Research, the National Institute for Health Research (NIHR) Cambridge Biomedical Research Centre, Cambridge Experimental Cancer Medicine Centre, Cambridge University Hospitals National Health Service Foundation Trust and the Cambridge Clinical Trials Unit (CCTU), and Addenbrooke's Charitable Trust. The views expressed are those of the authors and not necessarily those of the NHS, the MRC, NIHR or the Department of Health and Social Care.

**Competing interests:** The authors have read the journal's policy and have the following competing interests: ANP has received a speaker fee and travel expenses from GE Healthcare. SJW has received travel expenses from IPSEN. GDS has received educational grants from Pfizer, Astra Zeneca, and Intuitive Surgical; consultancy fees from Merck, Pfizer, EUSA Pharma and CMR Surgical; travel expenses and speaker fees from Pfizer. AW received a single Scientific Advisory Board fee from Roche. TE has received research support from Pfizer and AstraZeneca, and honoraria from Pfizer. TE was employed by AstraZeneca during part of the study duration and is now employed by Roche, and holds stock in AstraZeneca and Roche. Pfizer provided the study drug Sunitinib and an educational grant for the translational endpoint analysis, reported separately. There are no patents, products in development or marketed products associated with this research to declare. This does not alter our adherence to PLOS ONE policies on sharing data and materials.

## Conclusions

Physiological imaging confirmed efficacy of the anti-angiogenic agent 12 days after initiating therapy and demonstrated response to treatment. The change in diffusivity shortly after starting pre-surgical sunitinib correlated to PFS in mRCC undergoing nephrectomy, however, no parameter predicted OS.

## Trial registration

EudraCtNo: 2005-004502-82.

## Introduction

Renal Cell Carcinoma (RCC) is the most common malignant tumor of the kidney, with prognosis being highly stage-dependent. Despite the increasing detection of early-stage RCC as an incidental finding on cross-sectional imaging, 15–20% of patients still present with advanced, stage IV disease [1, 2]. 5-year overall survival (OS) exceeds 80% in patients diagnosed with stage I and II RCC, however, the prognosis of patients with locally advanced or metastatic RCC (mRCC) has a far lower 5-year OS at 11.3% [1, 2].

The therapeutic spectrum for advanced and metastatic RCC has expanded rapidly over the past decade. Whilst vascular endothelial growth factor (VEGF)-targeted therapy combined with an immune-checkpoint inhibitor has become the standard-of-care in fit patients with mRCC, (VEGF)-targeted therapy like sunitinib continues to be used in all lines of treatment [3, 4]. Expansion in the therapeutic options of mRCC requires a parallel improvement in methods of early detection of therapy response to optimize the time spent on effective therapy and reduce side-effects and costs in the absence of clinical benefit.

Current guidelines suggest two to four-monthly follow-up of mRCC patients on systemic therapy with contrast-enhanced CT for response assessment according to the Response Evaluation Criteria in Solid Tumors (RECIST 1.1) [5]. However, these criteria have well-known limitations, and there is no clinical evidence that RECIST-defined disease progression is a clinically valid endpoint that requires treatment interruption or modification [3]. Additionally, patterns of treatment response to immunotherapy and targeted therapy differ from cytotoxic chemotherapy where volume-effects predominate, and are insufficiently captured by RECIST at early time points [6]. While survival is correlated with volumetric response in mRCC, even stabilization of the disease is a valuable achievement, which is, however, difficult to detect morphologically [7, 8].

An early marker of response to VEGF-targeted treatment with tyrosine kinase inhibitors (TKI) would allow a switch to alternative anti-cancer therapy before the tumor burden increases in patients who derive no benefit from TKI. The mechanism of action of VEGF-targeted TKI like sunitinib, namely inhibition of tumor angiogenesis and vascular maintenance, supports the use of physiological MRI for the early detection of treatment response [9]. Intravoxel incoherent motion-type diffusion-weighted imaging (IVIM-DWI) can differentiate the contribution of microperfusion and tissue diffusion to the overall diffusion signal [10]. Meanwhile, blood oxygen level-dependent (BOLD) MRI is sensitive to tissue hypoxia, which has been shown to increase following sunitinib treatment in melanoma xenografts [11, 12]. Finally, tissue perfusion can be assessed using dynamic contrast-enhanced (DCE) MRI.

The safety and efficacy of 12 days of pre-surgical sunitinib were investigated in the NeoSun prospective clinical trial (EudraCT No: 2005-004502-82). Sunitinib was chosen as a well-investigated first-line agent in mRCC and for its potential as a neoadjuvant agent to reduce the

volume of the primary tumor [13]. This report investigates the effects of pre-surgical sunitinib therapy using quantitative, physiological and morphological MRI measurements and assesses whether these can be used to detect early treatment response and predict overall and progression-free survival in the NeoSun cohort.

## Materials and methods

### Participant selection

Imaging data from the NeoSun trial, a prospective, single-centre, single-arm phase II study of neoadjuvant sunitinib in mRCC, was assessed following a pre-planned analysis [14]. National research ethics committee approval was obtained (REC ref: 2005-004502-82) and written informed consent was provided by all study participants. Key inclusion criteria were histopathological confirmation of clear cell RCC with metastases judged by the treating clinician to potentially derive benefit from sunitinib prior to a planned cytoreductive nephrectomy. Participants were excluded if they had undergone previous treatment for mRCC or were unable to undergo MRI. Twenty-two consecutive participants were screened for inclusion in the trial between 04/2011 and 01/2014. Six participants were excluded prior to enrolment and four participants did not fulfil the criteria for study completion (see CONSORT diagram, Fig 1). This

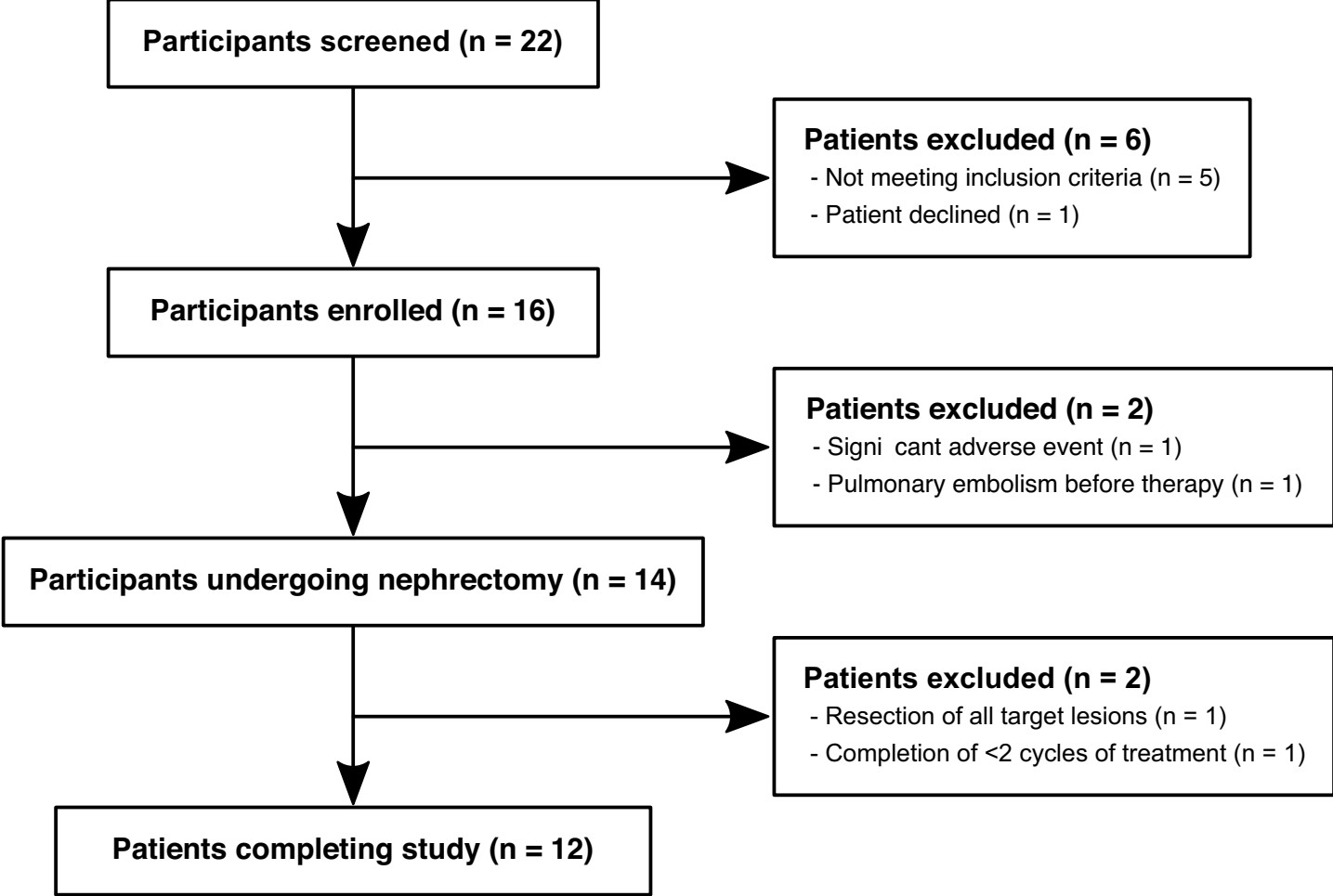

**Fig 1. CONSORT diagram explaining recruitment and reasons for exclusion of participants.**

manuscript reports the outcomes of the exploratory imaging endpoints of the NeoSun trial. Primary oncological outcomes of the trial have been published elsewhere [14] and only participants who completed the full MR-imaging protocol are included in this report. One participant did not tolerate the MRI. The incomplete dataset was of poor quality and, therefore, excluded from further analysis.

## Treatment

All participants underwent treatment with 50 mg sunitinib once daily for 12 days prior to radical cytoreductive nephrectomy; one participant due additionally underwent bilateral adrenalectomy for metastatic involvement. Sunitinib therapy was resumed after surgery on a repeating 6-week cycle (4 weeks-on, 2 weeks-off) until RECIST 1.1 defined disease progression was identified on 12-weekly follow-up CT, or patients experienced unacceptable toxicity, as assessed by the National Cancer Institute Common Terminology Criteria for Adverse Events, Version 4.0 [15].

## Survival

Overall survival (OS) was defined as the time between enrolment and the date of death from any cause. Participants who were lost to follow-up were censored at the last date when they were known to have been alive. Patients who were still alive on the 01/03/2020 were censored on this date, resulting in a median follow-up of 2.6 years (range: 0.6–8 years) for all patients and between 2.7 years and 8 years for surviving patients. Progression-free survival (PFS) was defined as the time between enrolment and the date of RECIST 1.1 defined progressive disease, or death from any cause.

## MRI technique

Multiparametric MRI was acquired at baseline and after 12 days of pre-surgical sunitinib treatment. Treatment response assessment was performed using contrast-enhanced CT of the thorax, abdomen and pelvis at baseline and every three months following nephrectomy, until disease progression.

MR imaging was performed on a 1.5 T Discovery MR450 system (GE Healthcare) using an 8-channel cardiac array coil. The multiparametric protocol included sagittal DWI using a dual-spin-echo echo-planar imaging sequence and b-values of 0, 150, 500, 700 and 900 s/mm$^2$ with TE 73.4 ms; TR 4000 ms; FoV 35x35 cm$^2$; slice thickness/gap 4/1 mm; acquisition matrix 104x104; 6 Nex; receiver bandwidth ±250 kHz; ASSET factor 2; scan time 5 minutes 16 seconds (Table 1). Trace-weighted images were acquired by averaging 3 orthogonal diffusion directions for the non-zero b-values. Saturation bands were used to reduce signals from outside the volume of interest. Although no specific DWI performance characterisations were performed for this specific study, the reliability of the body DWI measurements was carefully characterised as part of another study conducted on the same MRI system and during an overlapping time-period [16]. The S1 File provides detailed acquisition information for the remaining sequences. The median acquisition time was 61 minutes (table time, range 56–68 minutes). For the DCE-MRI acquisition, 0.1 mmol/kg of Gd-DOTA (Dotrarem, Guerbet) was used.

## Image processing

Analysis of the IVIM DWI data was performed using in-house developed software developed in MATLAB (The Mathworks, version R2010b). The four non-zero b-values were used for

**Table 1. Imaging parameters.**

| Sequence | TE / TR [ms] | Flip Angle [deg] | Matrix Size | Slice Thickness / Orientation | Other parameters |
|---|---|---|---|---|---|
| $T_2$w FRFSE | 48–69 / 4000 | 90 | 320 x 224 | 4 mm coronal | ETL: 10–13, RT |
| | | | | 1 mm gap | 2 Nex |
| $T_1$w FSPGR | 4.8 / 139 | 70 | 256 x 256 | 4 mm coronal | BH |
| | | | | 1 mm gap | 0.75 Nex, ASSET 2 |
| DW-EPI | 73.4/4000 | 90 | 104 x 104 | 4 mm sagittal | b-values: 0, 150, 500, 700 and 900 s/mm$^2$ |
| (IVIM type) | | | | 1 mm gap | 6 Nex, ASSET 2 |
| $R_2^*$ mapping | 4.76–47.6 (echo spacing 4.76) / 100 | 25 | 128 x 128 | 4 mm sagittal | BH |
| | | | | 1 mm gap | ASSET 2 |
| $T_1$ mapping | 1.6 / 3.9 | 1, 3, 5, 10, 15 and 20 | 160 x 160 | 5 mm coronal-oblique | BH |
| | | | | | 1 Nex |
| DCE-MRI | 1.6 / 3.9 | 18 | 160 x 160 | 5 mm coronal-oblique | intermittent bh, temporal resolution: 4.3–6.4 s |
| | | | | | 0.5 Nex |

TE: Echo Time, TR: Repetition Time, ETL: Echo Train Length, RT: respiratory triggered, BH: breath hold, DCE: dynamic contrast-enhanced, Nex: Number of excitations.

voxelwise mono-exponential calculations of the diffusivity $D$. The perfusion fraction $f_p$ was then estimated at a voxel level from the fractional difference between the measurements at b = 0 s/mm$^2$ and the extrapolated signal from the higher b-values. A repeatability coefficient of 20% was pre-defined as a threshold for the detection of treatment-related changes based on existing literature describing the repeatability of ADC measurements [17–19].

In-house software developed in MATLAB was used to generate $R_2^*$ and $T_2^*$ maps from the multi-echo gradient-echo images. The pixel-level data were fitted to a mono-exponential decay using the nonlinear Levenberg–Marquardt algorithm, and using a log-linear approximation to compute the initial values for the fits.

The $T_1$ mapping and DCE-MRI data were processed in MIStar (Apollo Medical Imaging Technology). Each dataset was pre-processed using a 3x3 median filter. The $T_1$ mapping and DCE datasets were co-registered both within and between the datasets to remove (as far as possible) spatial mis-registrations induced by motion. The Tofts model [20] using a population-averaged Arterial Input Function [21] was applied to calculate maps of the transfer constant $K^{trans}$ and quantify the contrast-concentration-versus-time curve for 90 seconds after the arrival of the contrast bolus (iAUC$_{90}$).

Regions-of-interest (ROI) were outlined for the normal-appearing ipsilateral kidney, the tumor and its necrotic/cystic parts by two radiologists with 8 (T.B.) and 5 (F.Z.) years of experience on the coronal $T_2$w FRFSE sequence, the sagittal ADC map, the sagittal $R_2^*$ and coronal-oblique $K^{trans}$ maps using ImageSetViewer Software, version 1.7 (University Health Network Toronto). The segmentation was undertaken blinded to the clinical outcome and RECIST measurements. The region of interest for the solid tumor components was generated from the subtraction of the necrotic/cystic parts from the entire tumor. Using custom-written MATLAB software, slice-wise ROIs were combined into whole volumes of interest and used to calculate the volumes of each tumor component (from $T_2$w imaging) and also the median values of $K^{trans}$, iAUC$_{90}$, $R_2^*$, $f_p$ and $D$ within the whole and solid tumor components for each participant. The test re-test variation in imaging parameters was compared in the normal renal tissue to appraise the repeatability of quantitative imaging measurements in tissues not targeted by sunitinib. A single radiologist (T.B.) performed the RECIST 1.1 assessment on the CT scans acquired at baseline and following nephrectomy. The contrast-enhanced CT scans

covererd the chest, abdomen and pelvis and included metastatic lesions which were not included in the field-of-view of the MRI scan which was restricted to the tumour-bearing kidney.

## Statistical methods

Statistical analysis was performed in the SAS suite (version 9.4). Differences before and after therapy were assessed with the non-parametric Wilcoxon signed-rank test. Differences in survival were visualised using Kaplan-Meier plots and survival distributions compared using log-rank tests. The Mann-Whitney U test was used for the comparison of unpaired samples. P values $\leq 0.05$ were considered as statistically significant.

## Results

The data of thirteen of the 22 participants screened for participation in this study was analyzed. Twelve patients (1 female, 11 male) were included in this study (Fig 1). The median age at enrolment was 60 years (range: 50–74, mean 61 years). The stage at nephrectomy was pT1b in one patient, pT3a in 10 patients and pT3b in one patient. Five patients were diagnosed with pN1 disease and all patients had cM1 disease. Participant characteristics are summarized in Table 2.

### Volumetric findings

At baseline, the mean volume of the primary tumor was 633 ml (SD ± 386 ml, range 175–1369 ml) of which 68% (15–95%) equivalent to 388 ml (± 249 ml, 104–903 ml,) was solid tumor tissue. Following pre-surgical treatment with sunitinib for 12 days, an average reduction of 25% to 507 ml (± 351 ml, 90–1213 ml) was observed in the total tumor volume (p < 0.001, *df* = 11,

**Table 2. Participant characteristics.**

| Characteristic | n = 12 |
|---|---|
| Age at enrolment (mean ± SD) | 61 ± 7.6 years |
| Gender (f/m) | 1 / 11 |
| Stage at nephrectomy | |
| pT1a | 0 |
| pT1b | 1 |
| pT2a | 0 |
| pT2b | 0 |
| pT3a | 10 |
| pT3b | 1 |
| pT4 | 0 |
| Location of metastasis at diagnosis | |
| Bone | 1 |
| Lung | 10 |
| Pleura | 2 |
| Adrenal | 2 |
| Lymph nodes | 5 |
| Pancreas | 1 |
| Overall Survival (median [range]) | 138 weeks [30–417] |
| Progression-free survival (median [range]) | 67 weeks [17–417] |

SD: Standard deviation.

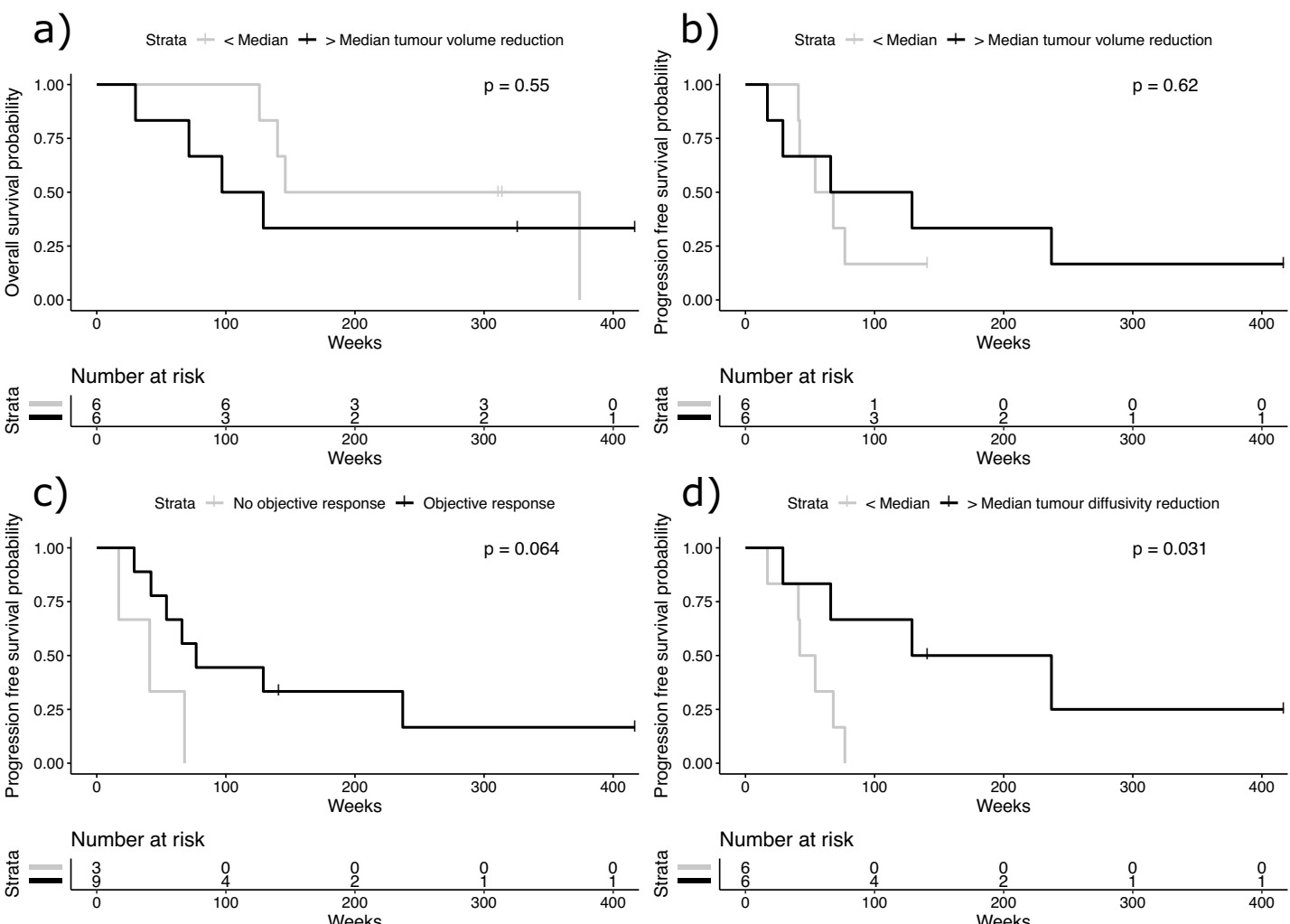

**Fig 2.** Kaplan Meier survival curves showing no overall survival benefit for participants with larger than median relative tumour volume reduction following 12 days of pre-surgical sunitinib therapy, however, there was a trend for participants with less volume reduction to survive longer (a). No difference in PFS was observed (b) Participants with a larger reduction in tumour diffusivity D experienced longer PFS (c) while there was a trend for participants with objective response according to RECIST 1.1 (partial or complete response) to show longer PFS than patients without objective response (stable or progressive disease) (d). P-values for log-rank test.

V = 78, previously reported [14]). Concurrently, the average volume of solid tissue decreased by 28% to 295 ml (± 217 ml, 81–749 ml), p < 0.001 (*df* = 11, V = 78), with the volume of necrotic/cystic tumor components only reducing by 17%, from 245 ml (± 263 ml, 10–885 ml) to 212 ml (± 251 ml, 8–830 ml), p = 0.005 (*df* = 11, V = 74). Furthermore, tumor necrosis was associated with decreased response to pre-surgical therapy with a negative correlation between the percentage of necrotic tumor volume at baseline and the overall reduction in tumor volume after 12 days of sunitinib therapy (Spearman rank correlation coefficient ρ = -0.69, 95%-CI: 0.10–0.97, p = 0.02). However, neither the reduction in whole tumor volume nor a solid tumor volume was associated with PFS and OS (Fig 2a/2b). The reduction in volume of the tumor was not associated with the overall RECIST response of the residual metastatic tumor burden following surgery (Fig 3A and 3B). There was a trend for objective response according to RECIST (partial or complete response as best RECIST response) being associated with longer PFS (p = 0.064, median PFS responders: 77 weeks, non-responders: 41 weeks, Fig 2C)

### Diffusion-weighted imaging

Following pre-surgical treatment, a reduction of the median diffusivity $D$ of the solid tumor by 7.7% from 1298 x $10^{-6}$ mm$^2$/s to 1200 x $10^{-6}$ mm$^2$/s was observed (p = 0.02, $df$ = 11, V = 68). Only two patients achieved a greater than 20% change in $D$, which was the a priori defined threshold for treatment effect in an individual patient. However, a greater than median reduction in solid tumor $D$ was associated with prolonged PFS (p = 0.031), median PFS 183 weeks vs. 48 weeks, Fig 2D). The baseline $D$ was not predictive of OS or PFS and the reduction in $D$ was not predictive of OS. Concurrently, the perfusion fraction of the solid tumor decreased by 19% from 0.24 to 0.19 (p < 0.001), however, a reduction in perfusion fraction exceeding the predetermined cut-off of 20% was not predictive of prolonged OS (p = 0.88).

The change in mean tumor diffusivity $D$ between baseline and follow-up showed a differential response between participants, with an 11% reduction in patients demonstrating an objective response according to RECIST (partial or complete response; n = 9), and a 3% increase for non-responders (n = 3);. Fig 3C The perfusion fraction $f_\mathrm{p}$ was not able to differentiate between responders and non-responders and neither $D$ nor $f_\mathrm{p}$ was associated with long-term treatment benefits.

### Blood oxygenation dependent imaging

The median $R_2^*$ from BOLD MRI increased within the solid parts of the tumor from 19 s$^{-1}$ to 28 s$^{-1}$ (p = 0.001, $df$ = 11). However, an $R_2^*$ at baseline above the median did not predict OS and PFS and the increase in R2$^*$ was not associated with improved response according to RECIST.

### T$_1$ map

A post-hoc analysis showed a median T1 of 1238 ms (interquartile range: 1184–1341) in the solid tumour at baseline which reduced to 1102 ms (994–1221 ms) after 12 days of sunitinib.

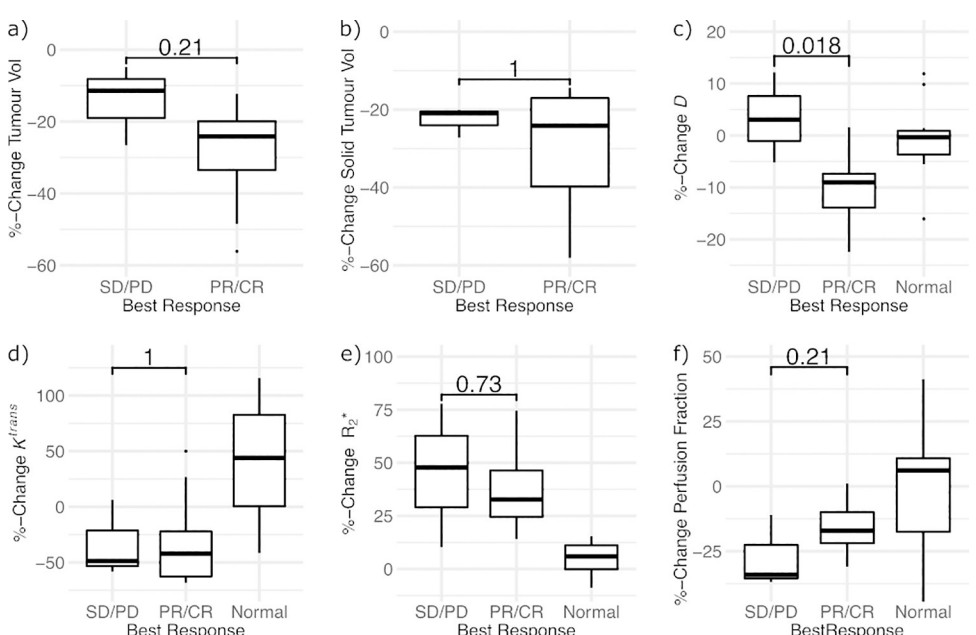

**Fig 3.** Treatment related changes: Percentage change in the whole tumour volume (a), solid tumour volume (b), solid tumour Diffusivity D (c), solid tumour $K^{trans}$ (d), solid tumour $R_2^*$ (e) and solid tumour perfusion fraction (f) between patients with stable/progressive disease (SD/PD) and partial/complete response (PR/CR). P-values for Mann-Whitney U test. *Relative changes in normal renal tissue illustrate the test re-test variability in imaging measurements.*

The median reduction by 6% was not significant (p = 0.094). Neither the tumour T1 at baseline nor its change following treatment was associated with the RECIST response or survival.

## Dynamic contrast-enhanced imaging

Median $K^{trans}$ and iAUC$_{90}$ both showed significant decreases within solid tumor components following treatment initiation, with a mean reduction in $K^{trans}$ from 0.415 min$^{-1}$ to 0.305 min$^{-1}$ (28% reduction, previously reported [14]), and of iAUC$_{90}$ from 0.55 to 0.36 mM min (38% reduction), respectively (p = 0.01 and 0.003, $df$ = 11, V = 71 and 78).

Nine participants had a reduction in $K^{trans}$ greater than the pre-determined repeatability coefficient of 20% which was selected based on existing literature [22, 23] and is in agreement with more recent findings [24]. However, these participants did not derive an overall or PFS benefit from this.

A typical example of the evolution of all physiological imaging parameters during the pre-surgical sunitinib treatment can be seen in Fig 4.

## Discussion

This study showed that early treatment response to pre-surgical sunitinib in RCC can be detected after 12 days using multiparametric MRI and that changes in diffusivity are associated with the volumetric response to treatment and progression-free survival. The data for this study was gathered in a prospective clinical trial where treatment and imaging were tightly controlled and the consistent timing of the follow-up MRI relative to the start of treatment was critical for assessing the early treatment response. Furthermore, the physiological parameters investigated in this study were functionally linked to the mechanism of action of sunitinib and could, therefore, report on the engagement of the drug with its target. The imaging data were acquired on a single MRI-scanner, eliminating inter-scanner variability as a confounding factor.

The reduction in tumor volume following sunitinib therapy was greatest in the solid tumor fraction. This may aid treatment planning with initial neo-adjuvant systemic therapy acting to debulk tumor volume in patients with large tumors, to reduce surgical challenges and risks. In these patients, the presence of large solid tumor components would suggest that a chemoreduction prior to cytoreductive surgery may be feasible with a short course of TKI therapy. Furthermore, the decrease in solid tumor volume could serve as a sensitive biomarker of treatment response. However, only the reduction in diffusivity was associated with PFS, with significant reductions in excess of 20% from the baseline values being observed in two patients. This result will need validation in future, larger cohorts and if confirmed, this imaging approach might be particularly relevant in localized disease where surgery is more impactful as a cure.

The changes observed in physiological imaging parameters were compatible with the expected mechanism of action of sunitinib. Consequently, these widely available MRI techniques could be useful surrogate markers for confirming successful engagement of anti-angiogenic agents with their biological target. Consistently, the perfusion fraction on IVIM-type DWI was reduced, representing a decrease in capillary volume; this was, however, not correlated with survival. This is expected given the decreased activation of the vascular endothelial growth factor receptors (VEGFR1 –VEGFR3) which leads to reduced angiogenesis and vascular maintenance [25]. Furthermore, the R$_2$* was increased, representing decreased oxygenation and vascularization, and the DCE transfer constant $K^{trans}$ and area under the contrast-enhancement curve (iAUC$_{90}$) were decreased, signifying decreased perfusion and vascular permeability following treatment. These imaging surrogates were in agreement with the decrease in immunohistochemical microvessel density observed in the same patient cohort [14].

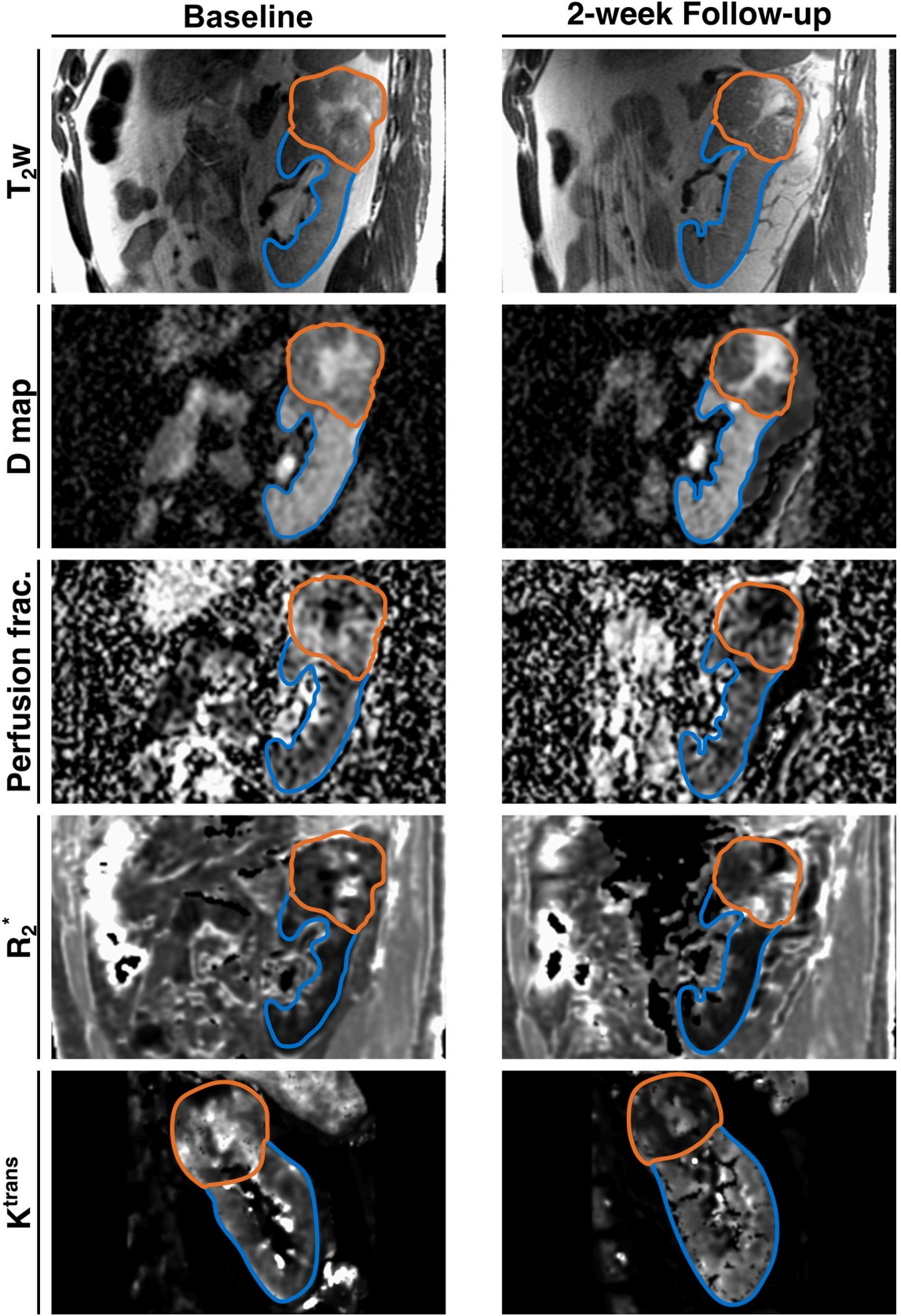

**Fig 4. Examples of the multiparametric MRI before and after 12 days of sunitinib therapy in a 52 year-old male patient.** The parametric maps before and after therapy are windowed equally. This patient showed an overall tumour volume reduction of 14%, with a decrease in volume of the solid tumour fraction by 33% and an increase in the necrotic/cystic tumour fraction of 8% after two weeks of therapy. Additionally, there was a reduction in diffusivity and perfusion fraction as well as an increase in $R_2^*$ in the solid tumour component. $K^{trans}$ decreased post-therapy, which is consistent with the mechanism of action of sunitinib and representative of most of the study patients. D map (diffusivity) and Perfusion fraction from IVIM-type DWI, $K^{trans}$: Contrast transfer constant, $R_2^*$: effective transverse relaxation rate, $T_2w$: $T_2$-weighted image.

Previous attempts at predicting treatment outcome in patients undergoing 15 days of sunitinib treatment for mRCC have shown an increased time-to-peak using contrast-enhanced ultrasound (CEUS). This was correlated with the RECIST response after 12 weeks, PFS and OS in a cohort of 38 patients [26]. However, follow-up results including a multi-center study involving 157 patients were not able to confirm these initial results [27, 28]. Contrary to these studies, sunitinib was the first-line treatment in all patients in NeoSun. MRI, as opposed to CEUS, has several advantages. It is operator-independent and the larger field-of-view allows imaging of the entire tumor volume and potentially multiple metastatic deposits in addition. This is particularly advantageous in mRCC where the response to TKI is known to show intra-patient heterogeneity [29]. Finally, CEUS is reliant on having a good acoustic window and little respiratory motion of the target lesion during the three-minute acquisition.

A larger post-treatment reduction in diffusivity of the solid component of the primary tumor was associated with a better RECIST response in the metastatic burden and longer PFS. The overall diffusion restriction increased, despite increasing histological necrosis which acts to increase ADC values [14]. Other studies have demonstrated increases in ADC values after three days of treatment which returned to baseline after ten days and remained unaltered after 3–4.5 months [30, 31]. In the current study, diffusivity was assessed in solid tumor components, while the others analyzed the entire tumor. Given the very heterogeneous nature of RCC, measuring the true diffusivity change following treatment is difficult. The assessment of the solid components alone is likely to be the most accurate measure of the response to therapy in the viable tumor. Furthermore, IVIM-type DWI provided information on diffusion restriction and relative capillary volume simultaneously, whereas previous studies investigated ADC, an aggregate of tissue diffusion and microperfusion [30, 31]. It is conceivable that these parameters undergo sequential changes as the tissue becomes edematous, necrotic, and is then restructured [31]. Furthremore, sunitinib is associated with increasing infiltration of tumour with CD8$^+$ T-cells, which may also lead to a reduction in diffusivity [32]. However, repeat diffusion-weight imaging at more frequent intervals would be needed to resolve these temporal changes.

This study has several limitations, including the small number of patients, however, this is similar to previous literature investigating multiparametric MRI in patients with RCC [33, 34]. The study was stopped ahead of its recruitment target due to slow participant accrual, consequently, the statistical power was decreased. As a single center trial, this will require independent validation to determine generalizability. Furthermore, recent results have shown that cytoreductive nephrectomy in metastatic RCC needs to be considered carefully and is most appropriate in patients with low-volume metastatic disease [3]. In particular, the correlation of imaging biomarkers with OS, a secondary aim of the NeoSun trial, should be investigated in an appropriately powered trial. Recent data has shown that mono-exponential fitting of $D_0$ should employ b-values $> 200$ s/mm$^2$ to avoid contributions from tissue perfusion [35]. As the data for NeoSun was acquired prior to this publication, the lowest b-value was 150 s/mm$^2$ and perfusion may have had a minor contribution to the overall signal.

Treatment monitoring on standard-of-care CT alone is challenging as tumor size is a late sign of response. In this study, the physiological parameters $K^{trans}$, $R_2^*$ and $f_p$, which are

sensitive to the antiangiogenic effect of TKIs, were significantly altered during early sunitinib treatment. Together with metabolic imaging and radiomics, they can serve as candidate features in future research. Further research on larger cohorts is required to determine the ability of early treatment response detected by MRI to guide treatment decision making and optimize the time-frame of effective therapy.

In conclusion, we demonstrate measurable changes in MRI features which are consistent with the proposed mechanism of action of multi-targeted receptor tyrosine kinase inhibitors, which were measurable after only 12 days of sunitinib therapy and related to the best response according to RECIST and PFS.

## Supporting information

**S1 File. Supporting methods.**
(DOCX)

**S2 File. Supporting results.**
(DOCX)

**S3 File. Data.**
(DOCX)

## Author Contributions

**Conceptualization:** Timothy Eisen, Sarah J. Welsh, Tristan Barrett.

**Data curation:** Fulvio Zaccagna, Sarah J. Welsh.

**Formal analysis:** Stephan Ursprung, Andrew N. Priest, Fulvio Zaccagna, Wendi Qian, Andrea Machin, Anne Y. Warren, Sarah J. Welsh, Tristan Barrett.

**Funding acquisition:** Timothy Eisen, Sarah J. Welsh.

**Investigation:** Stephan Ursprung, Andrew N. Priest, Fulvio Zaccagna, Anne Y. Warren, Timothy Eisen, Sarah J. Welsh, Tristan Barrett.

**Methodology:** Andrew N. Priest, Anne Y. Warren, Timothy Eisen, Sarah J. Welsh, Tristan Barrett.

**Software:** Andrew N. Priest, Tristan Barrett.

**Supervision:** Grant D. Stewart, Timothy Eisen, Sarah J. Welsh, Ferdia A. Gallagher, Tristan Barrett.

**Validation:** Wendi Qian, Andrea Machin, Ferdia A. Gallagher.

**Visualization:** Stephan Ursprung, Wendi Qian, Andrea Machin.

**Writing – original draft:** Stephan Ursprung.

**Writing – review & editing:** Andrew N. Priest, Fulvio Zaccagna, Wendi Qian, Andrea Machin, Grant D. Stewart, Anne Y. Warren, Timothy Eisen, Sarah J. Welsh, Ferdia A. Gallagher, Tristan Barrett.

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
