## [Decision Letter · Decision Letter 0]

1 Jul 2021

PONE-D-21-15997

Multiparametric MRI for assessment of early response to neoadjuvant sunitinib in renal cell carcinoma

PLOS ONE

Dear Dr. Barrett,

Thank you for submitting your manuscript to PLOS ONE. After careful consideration, we feel that it has merit but does not fully meet PLOS ONE’s publication criteria as it currently stands. Therefore, we invite you to submit a revised version of the manuscript that addresses the points raised during the review process.

ACADEMIC EDITOR:

This is a manuscript of potential interest. However, the Reviewers have raised numerous/major concerns and hence the paper is not acceptable in its present form for publication. Substantial revision along the lines recommended by the Reviewers may render the work suitable for publication. Because the issues raised by the review and requested revisions are rather extensive, my editorial decision will be deferred until completion of the next cycle of review. Please remember that a point by point response to each comment by the Reviewers is absolutely essential, since this is fair to the Reviewers who have put much effort/time into reviewing your paper, and it will definitely help to speed up the review process.

We look forward to receiving your revised manuscript.

Kind regards,

Marco Giannelli

Academic Editor

PLOS ONE

Additional Editor Comments:

I would like to note that MR scanner systems should be adequately characterized to obtain reliable quantitative diffusion-MRI measurements. The Authors should adequately discuss this issue, specifying whether they have performed any quality controls to assess the correct application of diffusion weighting gradients and non-linearity effects, which may be relevant for off-center acquisitions of kidneys as well as of the breast.

Journal Requirements:

3. We noted in your submission details that a portion of your manuscript may have been presented or published elsewhere.

[An abstract was presented at ASCO 2017, where drug safety and the duration of overall and progression-free survival were discussed. The abstract is attached to the submission.

A separate manuscript describing the survival outcome, drug safety and translational outcomes based on tissue analysis is currently under review. The only imaging parameters reported in this manuscript are tumour volume and Ktrans. The current, non-peer-reviewed version of the manuscript is attached to this submission.

In contrast to the above publications, the current submission focuses on the translational imaging analysis.]

4. Thank you for stating the following in the Financial Disclosure section:

[This study has received funding by Pfizer who provided the study drug Sunitinib and an educational grant for the translational endpoint analysis, reported separately.

The project was supported by Cancer Research UK (CRUK), the CRUK Cambridge Centre, Cambridge Trust, the CRUK & Engineering and Physical Sciences Research Council Cancer Imaging Centre in Cambridge and Manchester, the Mark Foundation for Cancer Research, the National Institute for Health Research (NIHR) Cambridge Biomedical Research Centre, Cambridge Experimental Cancer Medicine Centre, Cambridge University Hospitals National Health Service Foundation Trust and the Cambridge Clinical Trials Unit (CCTU), and Addenbrooke’s Charitable Trust.

Pfizer had a right to review the manuscript prior to publication. None of the funders had a role in the study design, data collection or analysis.]. 

We note that you received funding from a commercial source: Pfizer

Reviewers' comments:

Reviewer's Responses to Questions

**Comments to the Author**

1. Is the manuscript technically sound, and do the data support the conclusions?

Reviewer #1: Partly

Reviewer #2: Yes

2. Has the statistical analysis been performed appropriately and rigorously? 

Reviewer #1: Yes

Reviewer #2: No

3. Have the authors made all data underlying the findings in their manuscript fully available?

Reviewer #1: No

Reviewer #2: Yes

4. Is the manuscript presented in an intelligible fashion and written in standard English?

Reviewer #1: Yes

Reviewer #2: Yes

5. Review Comments to the Author

Reviewer #1: In this paper authors present a multi-parametric MRI approach to evaluate the response to sunitinib treatment in locally advanced renal cancer.

RECIST criteria, commonly used for evaluating response to therapy are based mainly on tumor size modifications and preset some limitations when immunotherapy or targeted therapy are administrated. Indeed sunitinib in renal cancer inhibits tumor angiogenesis that can early detect with multi-parametric MRI. Nevertheless the interesting and useful approach proposed by the authors, in general there are some issue concerning the technical aspects of this work. Furthermore few patients were included in this study.

In my opinion the study of the impact in terms of OS is not useful in this context. Indeed the primary goal of the study is to find quantitative MRI parameters to evaluate early tumor response. The impact of these on OS could be considered as a further step.

Materials and Methods

Why have you explained in details only DWI protocols while for the other techniques details are provided in table 1?

Lack of information about contrast agent for DCE in Materials and Methods.

What is the total acquisition time?

L142: Please insert Matlab version.

Before IVIM analysis, were images preprocessed for eddy current correction?

Authors state that for diffusion coefficient D calculation all the four non-zero b-bvalues DWI images were considered using a monoexponential fit. Why have you included also the b=150 s/mm2 for D calculation?

Is it well know that for b<200 s/mm2 we have also the contribution of perfusion in diffusion decay (please refers for example to Meeus et al. J. MAGN. RESON. IMAGING 2017;45:1325–1334.) for this reason it was proposed to calculate D including only bvalues higher that 200 s/mm2. Could you explain why have you implemented DWI analysis in this way?

The reference added for population-averaged AIF (ref n. 20) refers to abdomen. Can you argue if this population-averaged AIF can be applied to kidneys? How an altered kidney function could affect AIF?

Statistical methods

L170 Please insert SAS suite version.

Results:

L177: add the reference to fig 1 that provides the diagram with included/excluded patients. It is not clear in this section why 10 patients were excluded.

Reviewer #2: This is a study designed to develop an early imaging biomarker of metastatic RCC response to sunitinib. The use of a homogeneous study cohort, namely subjects with mRCC who all received neoadjuvant sunitinib as first-line therapy, is a significant strength. All scans were acquired on the same scanner, eliminating a potential source of variability. The acquisition of multiple quantitative MRI sequences is another strength, and this was done before and after 12 days of sunitinib treatment. The analysis of this rich image dataset has been done at the ROI/VOI level, rather than the pixel/voxel level, which would have been more powerful. One constraint clearly was that the sequences were acquired in different views (coronal, sagittal, oblique). Overall, this is an interesting study that is potentially useful. Some improvements to the quantitative image analytics would strengthen the report, as suggested below.

Specific Comments:

1) Reproducibility of tumor contouring: What was the Kappa for the two radiologists in defining solid and necrotic/cystic ROIs? An alternate to manual contouring would be to objectively define tumor sub-volumes using the histogram-based approach proposed by Chenevert, T. L. et al. Comparison of voxel-wise and histogram analyses of glioma ADC maps for prediction of early therapeutic change. Tomography 5, 7-14, 2019. https://doi.org/10.18383/j.tom.2018.00049

2) Reproducibility of DCE-MRI: Reference 21 pertains to brain MRI, which may be much more reproducible than MRI of abdominal tumors. And reference 22 pertains to reproducibility of mouse DCE-MRI, which doesn’t seem too relevant to clinical DCE-MRI. I would suggest using the clinical DCE-MRI reproducibility figures published by Galbraith et al. (NMR Biomed 2002 Apr;15(2):132-42. doi: 10.1002/nbm.731.) and Klaassen et al. (Magnetic Resonance Imaging 50:1-9, 2018, https://doi.org/10.1016/j.mri.2018.02.005).

3) Reproducibility of all quantitative MRI: In Figure 3, please also provide comparisons to corresponding changes in MRI parameters pre/post drug in tissues that are presumed to not be affected by drug. This is an option when repeat intra-session scanning was not done as part of the study. See, for example, Lorza, A.M.A., Ravi, H., Philip, R.C. et al. Dose–response assessment by quantitative MRI in a phase 1 clinical study of the anti-cancer vascular disrupting agent crolibulin. Sci Rep 10, 14449 (2020). https://doi.org/10.1038/s41598-020-71246-w.

4) Univariable analysis: Median values of Ktrans, iAUC90, R2*, fp and D within the whole and solid tumor components were analyzed. Was T1 (unenhanced) also evaluated in the same manner?

5) It appears that there were secondary lesions that were large enough for RECIST assessment. What were the post-sunitinib vs. pre-sunitinib quantitative MRI results (analogous to figure 3) from those additional tumors?

6) Discussion: Regarding the observed decrease in ADC of solid tumor following sunitinib treatment, is it known that the short-term response of tumor cells to sunitinib is cell swelling rather than apoptosis?

7) Did the authors perform a multivariable analysis of whether baseline R2*, fp , D, T1 (unenhanced), Ktrans, and iAUC90, taken together (or a subset), are predictive of post- vs. pre-sunitinib change in Ktrans or iAUC90? A change in Tofts model parameters with treatment would be more mechanistically-related to tumor-level drug pharmacodynamics than RECIST, which is a downstream patient-level response metric. Use of delta-DCEMRI rather than RECIST as ground truth for outcome would also allow for the primary plus any metastatic tumors to be analyzed, which would increase sample size. The input data could be augmented to include not just the median values of R2*, fp , D, T1 (unenhanced), Ktrans, and iAUC90, but also additional percentile values (e.g., 10th and 90th) to both improve the predictive performance of the model and to try and do away with the need to manually define “solid” and “necrotic/cystic” tumor regions. Even of the outcome metric remained RECIST, a multivariable analysis would still be useful.

Thank you for the opportunity to review your paper on MRI results from this nice study.

Natarajan Raghunand, PhD

6. PLOS authors have the option to publish the peer review history of their article (what does this mean?). If published, this will include your full peer review and any attached files.

Reviewer #1: No

Reviewer #2: **Yes: **Natarajan Raghunand, PhD

---

## [Author Response · Author response to Decision Letter 0]

1 Sep 2021

Reviewer #1: In this paper authors present a multi-parametric MRI approach to evaluate the response to sunitinib treatment in locally advanced renal cancer.

RECIST criteria, commonly used for evaluating response to therapy are based mainly on tumor size modifications and preset some limitations when immunotherapy or targeted therapy are administrated. Indeed sunitinib in renal cancer inhibits tumor angiogenesis that can early detect with multi-parametric MRI. Nevertheless the interesting and useful approach proposed by the authors, in general there are some issue concerning the technical aspects of this work. Furthermore few patients were included in this study.

In my opinion the study of the impact in terms of OS is not useful in this context. Indeed the primary goal of the study is to find quantitative MRI parameters to evaluate early tumor response. The impact of these on OS could be considered as a further step.

Thank you for this comment. The correlation of imaging markers with overall survival was defined as a secondary endpoint in the study protocol. It was included in the protocol since this information is clinically useful. We agree that appropriately powered studies would be required to investigate the relationship between imaging markers and survival further. We have included this as a limitation to our work. (Line 329)

Materials and Methods

1. Why have you explained in details only DWI protocols while for the other techniques details are provided in table 1?

Thank you very much for this question. We have chosen to detail the acquisition parameters in the main manuscript because the parameter which was found to be most useful (D0) is derived from this sequence. However, we have included similarly detailed descriptions of all the other sequences in the supplementary file S1 in addition to table 1. We have included a clarification in the methods section (Line 129).

2. Lack of information about contrast agent for DCE in Materials and Methods.

In this study, 0.1 mmol/kg of Gd-DOTA (Dotarem, Guerbet) was administered during the dynamic series. We have now included this information in the main manuscript in addition to the supplementary information. (Line 131)

3. What is the total acquisition time?

Thank you for raising this important point. The median total acquisition time was 61 minutes and the interquartile range extended from 56 to 68 minutes (n = 24 studies). We have included this information in the manuscript. (Line 130)

4. L142: Please insert Matlab version.

The MATLAB version used to process the data for this manuscript was R2010b, which has now been added to the manuscript. (Line 139)

5. Before IVIM analysis, were images preprocessed for eddy current correction?

Images were acquired using Dual Spin Echo EPI. This minimises eddy current effects so that eddy current correction was not needed.

6. Authors state that for diffusion coefficient D calculation all the four non-zero b-bvalues DWI images were considered using a monoexponential fit. Why have you included also the b=150 s/mm2 for D calculation?

Is it well know that for b<200 s/mm2 we have also the contribution of perfusion in diffusion decay (please refers for example to Meeus et al. J. MAGN. RESON. IMAGING 2017;45:1325–1334.) for this reason it was proposed to calculate D including only bvalues higher that 200 s/mm2. Could you explain why have you implemented DWI analysis in this way?

Thank you for raising this point. We agree that there remains a small perfusion contribution at b = 150 s/mm2 but the effect is much smaller than at lower b-values. This study was planned, and data acquired before the publication of the Meeus paper and therefore the b = 200 s/mm2 was not applied. We chose to neglect the small perfusion contribution at b = 150 s/mm2 and feel that this represents a reasonable compromise for this dataset. We have included this consideration in the limitations of our manuscript. (Line 330)

7. The reference added for population-averaged AIF (ref n. 20) refers to abdomen. Can you argue if this population-averaged AIF can be applied to kidneys? How an altered kidney function could affect AIF?

Thank you very much for this question. The effect of an altered renal function on the applicability of a population-averaged arterial input function will likely depend on the cause of a reduced renal function. A renal artery stenosis would affect both the renal function and the delivery of contrast-agent to the tumour. In contrast, primary or secondary glomerular diseases (e.g. focal segmental glomerulosclerosis, diabetic nephropathy) or obstructive nephropathy would result in a reduction in renal function with a maintained tumour perfusion. The population-averaged AIF described by Parker et al. is used commonly in the quantification of DCE-MRI of RCC 1,2.

Additionally, the primary endpoint of the NeoSun trial was a relative reduction in Ktrans. Therefore, even if the renal function was reduced in a way that had an effect on the contrast-enhancement of the tumour, the renal impairment would be unlikely to progress significantly within the 12-day baseline-to-follow-up period.

Importantly, adequate renal function was an inclusion criterion for the NeoSun trial. Therefore, none of the patients had a serum creatinine concentrations >1.5x the upper limit of the norm.

Statistical methods

8. L170 Please insert SAS suite version.

SAS suite version 9.4 was used for the analyses. This information has been added to the methods section of the manuscript. (Line 171)

Results:

9. L177: add the reference to fig 1 that provides the diagram with included/excluded patients. It is not clear in this section why 10 patients were excluded.

Thank you for this comment, we have added a reference to Figure 1 to line 177 (now line 178).

Reviewer #2: This is a study designed to develop an early imaging biomarker of metastatic RCC response to sunitinib. The use of a homogeneous study cohort, namely subjects with mRCC who all received neoadjuvant sunitinib as first-line therapy, is a significant strength. All scans were acquired on the same scanner, eliminating a potential source of variability. The acquisition of multiple quantitative MRI sequences is another strength, and this was done before and after 12 days of sunitinib treatment. The analysis of this rich image dataset has been done at the ROI/VOI level, rather than the pixel/voxel level, which would have been more powerful. One constraint clearly was that the sequences were acquired in different views (coronal, sagittal, oblique). Overall, this is an interesting study that is potentially useful. Some improvements to the quantitative image analytics would strengthen the report, as suggested below.

Specific Comments:

1) Reproducibility of tumor contouring: What was the Kappa for the two radiologists in defining solid and necrotic/cystic ROIs? An alternate to manual contouring would be to objectively define tumor sub-volumes using the histogram-based approach proposed by Chenevert, T. L. et al. Comparison of voxel-wise and histogram analyses of glioma ADC maps for prediction of early therapeutic change. Tomography 5, 7-14, 2019. https://doi.org/10.18383/j.tom.2018.00049

Thank you very much for this question. Each radiologist contoured either the T2w and DCE MRI sequence or the DWI and R2* map. As they were acquired in different orientations, the segmentations are not directly comparable. However, we have implemented an objective, histogram-based as described in the publication by Chenevert et al and included the results of the following analyses in the supporting materials (S1 for methods and S2 for results). We found that a threshold of 1.25x10-3mm2/s, as suggested by Chenevert et al., resulted in the identical interpretation of our results. While numerically different, the median D0 of the manually sub-segmented viable tumour and the automatically thresholded tissue were highly correlated (r = 0.75, p < 0.001, Figure R1). Similarly, responding tumours continued to show a significantly greater reduction in D0 following treatment and longer progression-free survival was associated with a greater reduction in D0.

 

a 

b 

Figure R1: a) D0 is strongly correlated between the manual sub-segmentation of viable tumour tissue and the automatic selection of viable voxels based on a published threshold. b) D0 remains a significant predictor of progression-free survival after the automated selection of viable tumour.

 

When the same thresholded masks were transferred to the perfusion fraction maps, a strong correlation with the manual sub-segmentation was observed (r = 0.90, p < 0.001). However, the perfusion fraction remained non-discriminatory between responders and non-responders and was not associated with PFS.

Figure R2: The perfusion fraction is strongly correlated between the manual sub-segmentation of viable tumour tissue and the automatic selection of viable voxels based on a published threshold for co-registered D0 maps.

No threshold for the differentiation of viable tumour and necrosis is available for Ktrans. We therefore chose to threshold on the goodness of fit of the extended Tofts model with a threshold of R2 = 0.5 to select only voxels which are perfused well enough to obtain a reasonable estimate of the kinetic parameters. We hypothesized that this would exclude necrotic and cystic tumour components. The Ktrans obtained through manual sub-segmentation of the tumour and the thresholding approach described above were strongly correlated (r = 0.69, p < 0.001). Furthermore, Ktrans remained non-discriminatory between responding and non-responding lesions and was not associated with survival.

Figure R3: Ktrans is strongly correlated between the manual sub-segmentation of viable tumour tissue and the automatic selection of viable voxels.

In the absence of a published threshold for the differentiation of viable tissue and necrosis based on R2*, we have employed the 90th percentile as an objective parameter in our analysis. Higher R2* values represent hypoxic tissue components. Similar to the median R2* in the viable tumour, the 90th percentile of the entire lesion increased significantly following treatment from 37 to 55 Hz (p < 0.001). However, neither the baseline value of R2* nor its change was associated with the RECIST response or survival. A comparison between the manual sub-segmentation and the thresholding at the 90th percentile showed a strong correlation between the two (r = 0.94, p < 0.001). 

Figure R4: R2* is strongly correlated between the manual sub-segmentation of viable tumour tissue and the automatic selection of viable voxels.

2) Reproducibility of DCE-MRI: Reference 21 pertains to brain MRI, which may be much more reproducible than MRI of abdominal tumors. And reference 22 pertains to reproducibility of mouse DCE-MRI, which doesn’t seem too relevant to clinical DCE-MRI. I would suggest using the clinical DCE-MRI reproducibility figures published by Galbraith et al. (NMR Biomed 2002 Apr;15(2):132-42. doi: 10.1002/nbm.731.) and Klaassen et al. (Magnetic Resonance Imaging 50:1-9, 2018, https://doi.org/10.1016/j.mri.2018.02.005).

Thank you very much for these suggestions. We have replaced the references accordingly. As the study by Klaassen et al. was published after the NeoSun trial had started, we have adapted the wording to reflect that Klaassen et al. could not have informed the choice of the threshold in NeoSun. (Line 250)

3) Reproducibility of all quantitative MRI: In Figure 3, please also provide comparisons to corresponding changes in MRI parameters pre/post drug in tissues that are presumed to not be affected by drug. This is an option when repeat intra-session scanning was not done as part of the study. See, for example, Lorza, A.M.A., Ravi, H., Philip, R.C. et al. Dose–response assessment by quantitative MRI in a phase 1 clinical study of the anti-cancer vascular disrupting agent crolibulin. Sci Rep 10, 14449 (2020). https://doi.org/10.1038/s41598-020-71246-w.

Thank you very much for this suggestion. We have included normal renal tissue in our analysis, similar to the method described by Lorza et.al. in their publication. The diffusivity D0 and the perfusion fraction were unchanged in the normal kidney following 12 days of sunitinib (p = 0.52 and 0.63, respectively). R2* was borderline significantly increased at the follow-up (p = 0.05) but the change was significantly lower than the one observed in tumours (p < 0.001). Finally, KTrans was significantly increased in the normal kidney following treatment (p = 0.03) while the transfer constant was decreased in tumour tissue. In summary, the test -re-test variability in the normal kidney does not fundamentally alter the interpretation of changes observed in tumours following treatment or differences between responders and non-responders.

Revised Figure 3: Percentage change in the whole tumour volume (a), solid tumour volume (b), solid tumour Diffusivity D (c), solid tumour Ktrans (d), solid tumour R2* (e) and solid tumour perfusion fraction (f) between patients with stable/progressive disease (SD/PD) and partial/complete response (PR/CR). P-values for Mann-Whitney U test. Relative changes in normal renal tissue illustrate the test re-test variability in imaging measurements.

We have chosen normal renal tissue for the assessment of the repeatability of imaging measures as it was included in the field-of-view regardless of the orientation of the image acquisition. Furthermore, it had more similar contrast-enhancement characteristics to the tumours than skeletal muscle and, unlike the liver, had a single blood supply. Even though sunitinib is considered a targeted treatment for clear cell renal cell carcinoma, it binds multiple targets including vascular endothelial growth-factor receptors, platelet-derived growth-factor receptors, c-KIT, the RET proto-oncogene, the granulocyte colony-stimulating factor receptor and CD135. The range of on- and off-target adverse effects of sunitinib highlight the broad spectrum of tissue in which the drug exerts an effect. Therefore, it is challenging to establish whether any normal tissue is unaffected by the drug.

4) Univariable analysis: Median values of Ktrans, iAUC90, R2*, fp and D within the whole and solid tumor components were analyzed. Was T1 (unenhanced) also evaluated in the same manner?

Thank you very much for this question. T1 maps were acquired as described in the supporting materials (S1). However, their analysis was not pre-specified in the statistical analysis plan of the NeoSun trial. A post-hoc analysis showed a median tumour T1 of 1238 ms (interquartile range: 1184 -1341) at baseline which reduced to 1102 ms (994 – 1221 ms) after 12 days of sunitinib. The median reduction by 6% was not significant (p = 0.094). Neither the tumour T1 at baseline nor its change following treatment was associated with the RECIST response or survival. (Line 239)

5) It appears that there were secondary lesions that were large enough for RECIST assessment. What were the post-sunitinib vs. pre-sunitinib quantitative MRI results (analogous to figure 3) from those additional tumors?

Thank you for this question. The RECIST assessment was performed on contrast enhanced CT of the chest, abdomen and pelvis which covered an extended field of view compared to the multiparametric MRI. The most frequent sites of metastasis of renal cancer are the lungs and bone. In this study, 10/12 patients harboured lung metastasis, four patients presented with enlarged mediastinal lymph nodes, one patient with pelvic bone metastasis and one with enlarged inguinal lymph nodes which were not covered by the field of view. One patient presented with a contralateral adrenal metastasis, one with a retrocaval lymph node metastasis and one with a metastasis in the pancreatic head which were not covered by the field of view in the sagittal acquisitions. We have included this clarification in the methods section of the manuscript. (Line 167)

6) Discussion: Regarding the observed decrease in ADC of solid tumor following sunitinib treatment, is it known that the short-term response of tumor cells to sunitinib is cell swelling rather than apoptosis?

Thank you for this question. There is only little information on very early physiological response mechanisms of clear cell renal cell carcinoma to sunitinib. The only study investigating sequential changes in ADC in this setting found decreasing diffusion restriction after three days of treatment which increased again after 10 days 3. The authors have also hypothesized that changing contributions from the development of oedema, cell swelling and necrosis may contribute to the temporal changes in tumour ADC. After 18 weeks, the ADC is unchanged relative to the baseline imaging in another study 4. Furthermore, sunitinib also increases tumour immune cell infiltration which may also increase the diffusion restriction 5. We have expanded on this in the discussion. (Line 319)

7) Did the authors perform a multivariable analysis of whether baseline R2*, fp , D, T1 (unenhanced), Ktrans, and iAUC90, taken together (or a subset), are predictive of post- vs. pre-sunitinib change in Ktrans or iAUC90? A change in Tofts model parameters with treatment would be more mechanistically-related to tumor-level drug pharmacodynamics than RECIST, which is a downstream patient-level response metric. Use of delta-DCEMRI rather than RECIST as ground truth for outcome would also allow for the primary plus any metastatic tumors to be analyzed, which would increase sample size. The input data could be augmented to include not just the median values of R2*, fp , D, T1 (unenhanced), Ktrans, and iAUC90, but also additional percentile values (e.g., 10th and 90th) to both improve the predictive performance of the model and to try and do away with the need to manually define “solid” and “necrotic/cystic” tumor regions. Even of the outcome metric remained RECIST, a multivariable analysis would still be useful.

Thank you very much for this question. 

We agree with the reviewer that the inclusion of objective parameters in addition to the manual sub-segmentation specified in the trial protocol would be useful. Therefore, we have included the threshold approach described by Klaassen et al. for the D0 and perfusion fraction maps. We have furthermore derived objective Ktrans measures through thresholding of the goodness of fit of the Toft’s model at R2 > 0.50, this only included voxels in the final analysis where kinetic parameters could be estimated with reasonable certainty and excluded poorly enhancing cystic and necrotic areas. The 90th percentile of the R2* distribution was selected as no threshold to distinguish viable and necrotic tissue has been published. The 90th percentile represents the most hypoxic tissue components. Please refer to question 3 of reviewer 2 for the results of these analyses.

We have not assessed whether any of the parameters was predictive of the reduction in Ktrans or iAUC90 because this was not specified in the trials statistical analysis plan. A post-hoc analysis showed that only the baseline Ktrans and iAUC90 were predictive of the relative reduction (p = 0.02 and 0.008, respectively). A higher Ktrans and iAUC90 at baseline were predictive of a greater reduction in Ktrans (R2adj = 0.35 and 0.64, respectively). Response prediction aims to anticipate patient benefit. Prolonged progression-free survival and RECIST are widely used surrogate markers of treatment benefit. The relationship between the pharmacodynamically relevant reduction in Ktrans and patient outcome is less well established. We have therefore chosen to summarise these results in the supplementary materials.

We have not performed a multi-variate analysis of all or a combination of the parameters. The reason was that the study was not powered for the development of such a multi-factorial model. As described in response to comment 5, metastatic deposits were outside of the field of view of the MRI acquisition and could unfortunately not serve to increase the sample size. Furthermore, using multiple lesions from one patient would introduce dependencies among data points which may lead to a less generalizable model. Overall, we felt that a population of 12 patients was not sufficient to explain a multifactorial model.

Thank you for the opportunity to review your paper on MRI results from this nice study.

Natarajan Raghunand, PhD

References

1. Wang HY, Su ZH, Xu X, et al. Dynamic Contrast-enhanced MR Imaging in Renal Cell Carcinoma: Reproducibility of Histogram Analysis on Pharmacokinetic Parameters. Sci Rep. 2016;6. doi:10.1038/srep29146

2. Wang H, Su Z, Ye H, et al. Reproducibility of Dynamic Contrast-Enhanced MRI in Renal Cell Carcinoma. Medicine (Baltimore). 2015;94(37):e1529. doi:10.1097/MD.0000000000001529

3. Desar IME, Voert EGW ter, Hambrock T, et al. Functional MRI techniques demonstrate early vascular changes in renal cell cancer patients treated with sunitinib: a pilot study. Cancer Imaging. 2011;11(1):259. doi:10.1102/1470-7330.2011.0032

4. Bharwani N, Miquel ME, Powles T, et al. Diffusion-weighted and multiphase contrast-enhanced MRI as surrogate markers of response to neoadjuvant sunitinib in metastatic renal cell carcinoma. Br J Cancer. 2014;110(3):616-624. doi:10.1038/bjc.2013.790

5. Haywood S, Chen R, Pavicic P, et al. Sunitinib’s effect on tumor infiltration of CD8 T cells in renal cell carcinoma (RCC) and modulation of their function by altering VEGF-induced upregulation of PD1 expression. https://doi.org/101200/jco2016342_suppl591. 2016;34(2_suppl):591-591. doi:10.1200/JCO.2016.34.2_SUPPL.591

---

## [Decision Letter · Decision Letter 1]

11 Oct 2021

Multiparametric MRI for assessment of early response to neoadjuvant sunitinib in renal cell carcinoma

PONE-D-21-15997R1

Dear Dr. Barrett,

We’re pleased to inform you that your manuscript has been judged scientifically suitable for publication and will be formally accepted for publication once it meets all outstanding technical requirements.

Kind regards,

Marco Giannelli

Academic Editor

PLOS ONE

Additional Editor Comments (optional):

Reviewers' comments:

Reviewer's Responses to Questions

**Comments to the Author**

1. If the authors have adequately addressed your comments raised in a previous round of review and you feel that this manuscript is now acceptable for publication, you may indicate that here to bypass the “Comments to the Author” section, enter your conflict of interest statement in the “Confidential to Editor” section, and submit your "Accept" recommendation.

Reviewer #1: All comments have been addressed

Reviewer #2: All comments have been addressed

2. Is the manuscript technically sound, and do the data support the conclusions?

Reviewer #1: Yes

Reviewer #2: Yes

3. Has the statistical analysis been performed appropriately and rigorously? 

Reviewer #1: Yes

Reviewer #2: Yes

4. Have the authors made all data underlying the findings in their manuscript fully available?

Reviewer #1: Yes

Reviewer #2: Yes

5. Is the manuscript presented in an intelligible fashion and written in standard English?

Reviewer #1: Yes

Reviewer #2: Yes

6. Review Comments to the Author

Reviewer #1: The authors have addressed all the issues underlined by the reviewers. In my opinion the manuscript in this reviewed version it is worth for publication

Reviewer #2: (No Response)

7. PLOS authors have the option to publish the peer review history of their article (what does this mean?). If published, this will include your full peer review and any attached files.

Reviewer #1: No

Reviewer #2: No

---

## [Editor Report · Acceptance letter]

15 Oct 2021

PONE-D-21-15997R1 

Multiparametric MRI for assessment of early response to neoadjuvant sunitinib in renal cell carcinoma 

Dear Dr. Barrett:

I'm pleased to inform you that your manuscript has been deemed suitable for publication in PLOS ONE. Congratulations! Your manuscript is now with our production department. 

Kind regards, 

on behalf of

Dr. Marco Giannelli 

Academic Editor

PLOS ONE